# A Two-Tier Perspective on Inference-Time Parallelism in Multi-Agent LLM Systems

Zihan Xu [* 1]   Haolin Tian [* 1]   Hai Jiang [1]

## Abstract

Large language model (LLM)-driven multi-agent systems typically require multiple model invocations and complex coordination during inference, and their execution strategies directly affect system accuracy, latency, and computational cost. Parallel execution provides a means to improve inference-time efficiency. From the perspective of inference-time execution, this paper models parallelism in multi-agent systems as two distinct levels of decision processes: Replica Parallelism, which explores multiple complete solution paths at the task level, and Structural Parallelism, which enables concurrent execution within a single solution path through task decomposition. However, the roles of different forms of parallelism and their interrelationships still lack systematic study in terms of unified organization and coordination. We therefore propose TIPEX, a controllable execution framework that unifies these two levels of parallelism and coordinates their roles within the inference process under a unified execution semantics while supporting systematic combinations and analyses of different parallel strategies and parameter configurations. Systematic experiments on the GAIA benchmark demonstrate that inference-time parallelism can significantly improve accuracy and reduce end-to-end latency at the cost of increased token consumption. Further analysis shows that Replica and Structural Parallelism exhibit complementary effects across task complexities, with tasks of intermediate difficulty benefiting most from their coordination, while overly aggressive parallel strategies do not necessarily yield better performance.

*Equal contribution   [1]Beijing University of Posts and Telecommunications.   Correspondence to: Hai Jiang <hai.jiang@bupt.edu.cn>.

*Proceedings of the 43rd International Conference on Machine Learning*, Seoul, South Korea. PMLR 306, 2026. Copyright 2026 by the author(s).

## 1. Introduction

The rapid advancement of large language models (LLMs) has spurred growing interest in multi-agent systems for complex tasks (Li et al., 2023; Wu et al., 2024; Chen et al., 2024; Hong et al., 2024; Fourney et al., 2024). By composing agents with different roles and capabilities, these systems support problem decomposition, tool use, information gathering, and collaborative execution. In practice, however, their inference-time behavior is often dominated by serial coordination: agents repeatedly call LLMs and tools, exchange intermediate states, and append new evidence into a growing context. As the number of execution rounds increases, this serial process can create long critical paths, context expansion and dilution, error propagation, and high wall-clock latency, all of which directly affect system accuracy, efficiency, and cost.

Parallel execution provides a natural way to mitigate these bottlenecks. Existing systems already instantiate this idea in different forms, such as running multiple agent teams or solution attempts in parallel, decomposing a task into subagents, or constructing task graphs for parallel scheduling (Zhang et al., 2025a; Kim et al., 2024; Zhang et al., 2025b; Yu et al., 2025). These approaches demonstrate that parallelism can improve multi-agent execution, but they usually study a particular portion of the design space in isolation. Consequently, it remains unclear how different forms of parallelism relate to one another, what roles they play in the accuracy–latency–cost trade-off, and whether combining them yields complementary or conflicting effects.

Motivated by this gap, we view inference-time parallelism as a two-tier design space. We distinguish *Replica Parallelism*, which explores multiple complete solution paths at the task level to expand solution-space coverage, from *Structural Parallelism*, which exposes concurrent execution opportunities within a single solution path by decomposing and scheduling agent and tool calls. Replica Parallelism primarily trades additional computation for robustness and accuracy, whereas Structural Parallelism primarily compresses the execution critical path to reduce wall-clock latency. Because both tiers consume shared resources and jointly determine the final output, they cannot be understood only as independent optimizations.

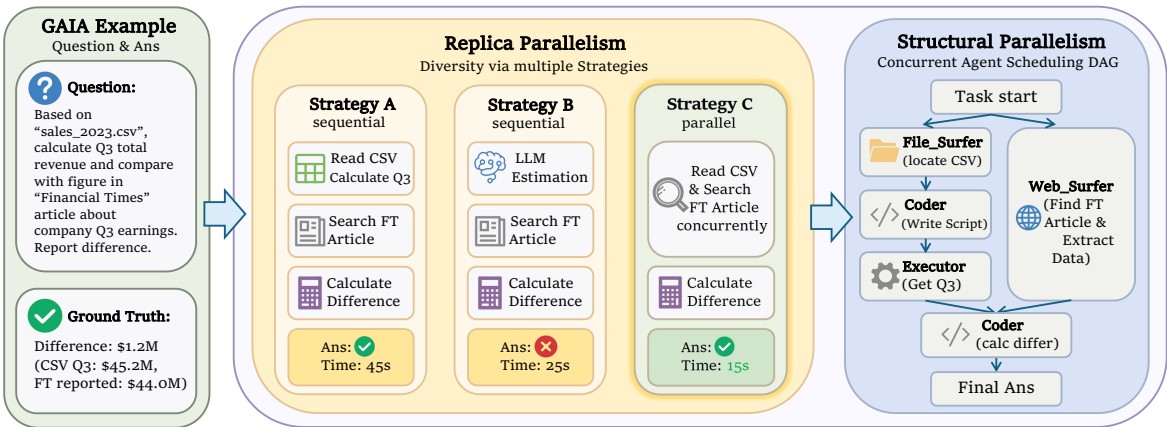

*Figure 1.* A running example illustrating inference-time parallelism in multi-agent systems. The example shows how Replica Parallelism explores multiple complete solution paths in parallel, while Structural Parallelism enables concurrent execution of agent and tool calls within a single solution path.

Based on this perspective, we propose **TIPEX** (**T**wo-tier **I**nference-time **P**arallel **EX**ecution), a controllable execution framework that unifies Replica Parallelism and Structural Parallelism for multi-agent systems. As Figure 1 shows, TIPEX separates the hierarchy and responsibilities of the two decision layers while supporting flexible combinations of generation, scheduling, and selection strategies under a unified execution semantics. This makes it possible to compare configurations, analyze interaction effects, and characterize when additional parallelism is useful or harmful while keeping the underlying agent pipeline unchanged.

Using TIPEX, we conduct empirical studies on inference-time parallel execution across tasks of varying difficulty and resource constraints. On GAIA (Mialon et al., 2024), inference-time parallelism improves accuracy and reduces end-to-end latency at the cost of increased token consumption. Our fine-grained analyses further show that the two tiers interact non-additively: moderate Replica Parallelism combined with Balanced Structural Parallelism gives the most robust trade-off, whereas overly aggressive configurations may introduce redundant execution and degrade reasoning quality. These results reveal clear applicability boundaries and motivate task-aware parallel execution. In summary, our contributions are:

- We model inference-time parallelism as two levels of decision making—Replica Parallelism and Structural Parallelism—and treat it as a structured design space.

- We propose TIPEX, a controllable execution framework that unifies and jointly orchestrates two-tier parallelism for multi-agent systems.

- TIPEX enables systematic strategy comparison and empirical characterization of accuracy–latency–cost trade-offs and applicability boundaries.

## 2. Related Work

### 2.1. LLM-Based Multi-Agent Systems

Recent advances in large language models have enabled a new class of LLM-based multi-agent systems that address complex, long-horizon tasks through coordinated reasoning and execution. Instead of relying on a single monolithic agent, these systems decompose tasks across multiple role-specialized agents that interact via dialogue, delegate subtasks, and iteratively refine intermediate results. Early paradigm-level studies focus on modeling communicative collaboration. CAMEL formalizes agent cooperation through role-playing and structured dialogue protocols (Li et al., 2023), demonstrating how explicit interaction schemes can guide coordinated problem solving. Complementary to this line, engineering-oriented frameworks emphasize reusable orchestration and tool integration. AutoGen offers a general-purpose programming framework for multi-agent conversations (Wu et al., 2024), AgentVerse provides a systematized organization for collaborative agents and emergent behaviors (Chen et al., 2024), and MetaGPT introduces standardized operating procedures to align multi-role teams toward consistent outputs (Hong et al., 2024). Beyond pre-defined workflows, Magentic-One targets open-ended tasks by employing a centralized orchestrator to coordinate multiple specialized agents. It supports multi-step interactions over the web, files, and code, and reports end-to-end performance across several multi-agent benchmarks (Fourney et al., 2024). Taken together, these works suggest that the effectiveness of multi-agent systems is not solely determined by model capacity, but is fundamentally constrained by the structure of their inference-time execution and coordination processes, motivating a closer examination of how such execution structures are organized and how they shape system-level performance trade-offs.

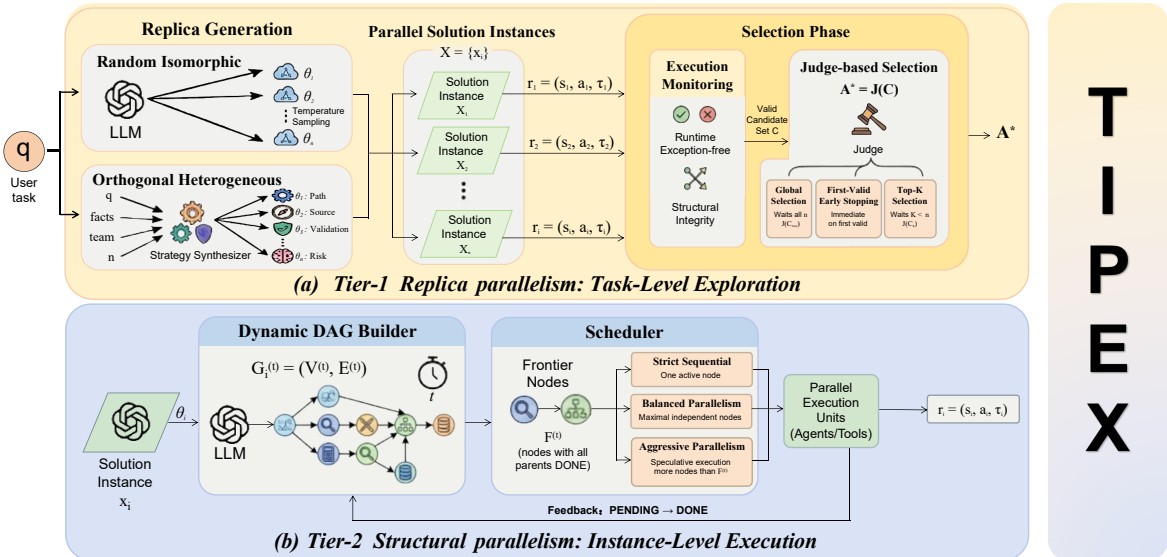

*Figure 2.* Overview of **TIPEX**. TIPEX organizes inference-time parallelism into two coordinated tiers. **Tier-1** applies Replica Parallelism for task-level solution exploration, including solution generation and selection. **Tier-2** applies Structural Parallelism for instance-level execution by dynamically constructing a dependency-aware DAG and scheduling agent and tool calls in parallel.

## 2.2. Inference-Time Parallelism in Multi-Agent LLM Systems

Inference-time latency has emerged as a critical bottleneck in multi-agent LLM systems, motivating a growing body of work on parallel execution to alleviate the cost of strictly sequential reasoning. Despite their diversity, existing approaches largely fall into two paradigms. The first introduces parallelism at the task level by concurrently exploring multiple solution trajectories or executing multiple agent teams, trading computational resources for improved accuracy or reduced end-to-end latency. M1-Parallel, for example, runs multiple teams in parallel and integrates alternative decision strategies to balance latency and performance (Zhang et al., 2025a). The second paradigm focuses on intra-trajectory structured parallelism, where a single reasoning process is decomposed into concurrent subtasks or function calls and scheduled in a dependency-aware manner during inference. LLMCompiler exemplifies this line by combining planning, dispatching, and execution to enable parallel function calls, thereby reducing latency and computation cost (Kim et al., 2024). Building on this idea, several recent systems adopt task graphs as intermediate representations for deriving parallel execution schedules. Plan-over-Graph constructs a task graph before generating a parallel plan (Zhang et al., 2025b), while DynTaskMAS leverages dynamic task graphs to drive asynchronous and parallel multi-agent execution (Yu et al., 2025). While these works demonstrate the benefits of individual parallelization strategies, they treat each form of parallelism in isolation. A unified analysis of their hierarchical roles, compositional patterns, and mutual interactions remains missing.

## 3. Method

### 3.1. Framework Overview

We formalize **TIPEX** as a hierarchical decision-making framework that separates inference-time reasoning into two decoupled but complementary levels of parallel control, each operating at a distinct granularity.

At the outer level, **Replica Parallelism** governs exploration across alternative solution hypotheses. It instantiates and manages a set of mutually independent solution replicas at the task level, aiming to increase solution robustness or reduce latency through parallel exploration of diverse reasoning trajectories.

At the inner level, **Structural Parallelism** governs execution within each solution replica. It translates an abstract solution into a structured execution graph and exploits conditional independence among reasoning steps by scheduling agent interactions and tool invocations concurrently whenever dependency constraints allow.

Formally, given a user query $q$, the solution orchestration layer initializes a set of $n$ solution replicas $\mathcal{X} = \{x_i\}_{i=1}^{n}$ via a stochastic generation process

$$x_i \sim \Phi(q, \theta_i), \quad i = 1, \ldots, n,$$

where $\Phi$ denotes an LLM-driven generative policy and $\theta_i$ parameterizes the $i$-th solver instance, enabling controlled heterogeneity across replicas.

Each solution replica $x_i$ is executed as a sequential decision

process and yields an outcome tuple

$$r_i = (s_i, a_i, \tau_i),$$

where $s_i \in \{0, 1\}$ indicates task success as determined by a verifier, $a_i$ denotes the produced answer when successful, and $\tau_i$ captures the complete execution trace.

A **solution orchestrator** serves as a meta-level controller that observes the evolving set of instance outcomes $\{r_i\}$. According to a predefined selection strategy, it determines when to terminate parallel exploration and aggregates the completed replicas to produce the final system output $A^*$ (Section 3.2).

Within each solution replica, execution is managed by an **execution scheduler** that maintains a dynamic directed acyclic graph $G_i = (V_i, E_i)$, where nodes represent computational units such as LLM inferences or tool calls, and edges encode data dependencies and temporal constraints (Section 3.3). The overview of TIPEX is shown in Figure 2.

## 3.2. Replica Parallelism

Replica Parallelism addresses uncertainty in LLM-based reasoning by parallelizing solution exploration at the task level. It decomposes the process into two stages: a *solution generation* stage that instantiates multiple, potentially diverse solution hypotheses, and a *solution selection* stage that aggregates their execution outcomes to determine the final system output.

### 3.2.1. SOLUTION GENERATION

We distinguish two solution generation strategies based on how configuration parameters $\theta_i$ are constructed.

**Random Isomorphic Generation (RIG).** All solution replicas share the same generation instruction, i.e., $\theta_1 = \cdots = \theta_n$. Diversity arises solely from stochastic decoding:

$$x_i \overset{\text{i.i.d.}}{\sim} \Phi(q, \theta), \quad T > 0.$$

This strategy generalizes self-consistency (Wang et al., 2022) by drawing multiple independent samples to approximate dominant reasoning modes in the model's distribution, thereby reducing errors from individual trajectories.

**Orthogonal Heterogeneous Generation (OHG).** We further propose an Orthogonal Heterogeneous Generation strategy grounded in meta-reasoning. Here "orthogonal" means methodologically diverse rather than mathematically orthogonal. Rather than relying solely on stochastic decoding, the system explicitly synthesizes a set of semantically distinct solution strategies:

$$\{\theta_1, \ldots, \theta_n\} \leftarrow \text{Synthesize}(q, \text{facts}, \text{team}, n).$$

The synthesis process promotes diversity along dimensions such as solution pathway, information source preference, verification mechanism, and risk preference. As a result, parallel instances differ not merely in random seeds, but in their underlying problem-solving logic, yielding structured complementarity across replicas. Appendix A provides additional diversity diagnostics.

### 3.2.2. SOLUTION SELECTION

The goal of solution selection is to determine the final system output $A^*$ from completed replicas. We model this process as a discriminator-based two-stage procedure consisting of *validity filtering* and *comparative selection*.

First, a replica is considered a valid candidate if it completes without runtime exceptions and produces a well-formed final answer and execution trace. Invalid instances are excluded from further consideration.

Given the candidate set $\mathcal{C}$, a judge operator $\mathcal{J}$ assigns each candidate a score $S(a_i, \tau_i \mid q)$ by combining LLM-based semantic assessment with deterministic checks. The judge is implemented as a standalone LLM-based evaluator that scores answer definiteness, evidence strength, and reasoning consistency, with the full rubric and overhead reported in Appendix A. The final output is selected as

$$x^* = \arg \max_{x_i \in \mathcal{C}} S(a_i, \tau_i \mid q).$$

We instantiate three orchestration strategies. **Global Selection (GS)** waits until all replicas complete and then applies $\mathcal{J}$ to the full candidate set. **First-Valid Early Stopping (FVES)** returns the first completed replica that passes validity filtering. **Top-$K$ Selection (TKS)** waits until $k$ valid candidates are available, then applies $\mathcal{J}$ only to this subset.

## 3.3. Structural Parallelism

Structural Parallelism aims to reduce end-to-end latency within a single solution replica by exploiting parallelism among dynamically generated subtasks. Since agent execution unfolds online and depends on intermediate reasoning outcomes, we model execution as dynamic DAG expansion and scheduling under partial observability.

Each solution replica maintains a monotonically growing directed acyclic graph $G_i^{(t)} = (V^{(t)}, E^{(t)})$, where nodes correspond to atomic computation units and edges encode strict data or logical dependencies. At each time step, the scheduler identifies the executable frontier

$$F(t) = \{v \in V(t) \mid v \text{ is causally ready at } t\}.$$

and dispatches tasks according to one of three policies:

- **Strict Sequential (SS)** organizes structural execution in the original agent trajectory order, where subtasks are processed step by step without structural concurrency.

- **Balanced Parallelism (BP)** organizes structural parallelism at a moderate granularity, executing naturally separable subtasks in parallel while maintaining the original high-level task decomposition.

- **Aggressive Parallelism (AP)** organizes structural parallelism at a finer granularity, decomposing the workflow into more parallelizable subtasks to expose a larger concurrent execution space.

Node-level failures are handled locally with bounded retries and dependency-aware continuation: failed hard dependencies terminate the affected chain, while independent branches continue along the available DAG frontier. Additional implementation details are given in Appendix A.

## 4. Experiments

### 4.1. Experimental Setup

**Dataset and Tasks.** To comprehensively evaluate the effectiveness of TIPEX in realistic and complex scenarios, we adopt the GAIA benchmark (General AI Assistants benchmark) (Mialon et al., 2024) as our primary testbed. GAIA covers a diverse set of task modalities, including information retrieval, multimodal reasoning, code execution, and web browsing. In addition, GAIA categorizes tasks into three increasing difficulty levels (Level 1–3), which provides a natural stratification for analyzing the behavior of inference-time parallel execution under different task complexities and reasoning depths.

**Baseline.** We choose Magentic-One (Fourney et al., 2024) as the baseline, a highly robust multi-agent orchestration framework. Notably, TIPEX is implemented directly on top of the Magentic-One architecture. Except for the introduction of Replica Parallelism and Structural Parallelism, all other system configurations are kept strictly identical to the baseline, ensuring fair and controlled comparisons.

**Evaluation Metrics.** We evaluate system performance from three perspectives: (1) **Accuracy**, which measures whether a task is successfully completed. To decouple generation quality from selection quality when evaluating solution selection mechanisms, we introduce **Oracle Accuracy**. This metric is defined as successful if at least one correct solution exists in the set of $n$ parallel candidates $\mathcal{C}$, representing the upper bound performance under an ideal (perfect) judge. (2) **Wall-clock Time**, which measures inference-time efficiency by recording the absolute physical time from task initiation to the final response. (3) **Token Consumption**, which quantifies the computational cost incurred during inference and is used to analyze the overhead introduced by parallel execution.

**Implementation Details.** All experiments are conducted on a server equipped with $4 \times$ Intel(R) Xeon(R) Platinum 8352V CPUs (@ 2.10GHz). To ensure reproducibility and accurate evaluation of concurrency effects, we implement the system using Python asynchronous I/O with a high-concurrency event loop, ensuring that latency gains from Structural Parallelism arise from algorithmic parallelization rather than blocking I/O artifacts. Unless otherwise specified, we adopt **Balanced Parallelism** as the default Structural Parallelism strategy, **Orthogonal Heterogeneous Generation** as the solution generation strategy, set the number of solution instances to $n = 3$, and use the **Top-$K$ Selection** strategy with $k = \lceil 3/2 \rceil = 2$. The rationale for these default settings is discussed later. We use Qwen-Plus (Yang et al., 2025) as the default model. Although Magentic-One already exhibits strong robustness, given the high reasoning complexity of GAIA tasks, we apply stratified repetition to obtain more reliable estimates: Level 1 and Level 2 tasks are repeated three times, while Level 3 tasks are repeated five times.

### 4.2. Main Results

Table 1 presents the main evaluation results. Our findings can be summarized as follows.

**Inference-time parallelism trades increased token consumption for improved accuracy and reduced latency.** Compared to Magentic-One, parallel configurations achieve higher accuracy and lower end-to-end latency across all three difficulty levels. For example, accuracy improves from approximately 43% to 52–57% on Level 1 tasks, and from around 26% to up to 39% on Level 2 tasks. Meanwhile, execution time is reduced by roughly 15%–40% in most configurations. These gains, however, are accompanied by a substantial increase in token consumption.

**The combination of OHG and TKS achieves the best overall trade-off.** Among different strategy combinations, OHG combined with TKS yields the most balanced performance. OHG with GS achieves high accuracy but incurs higher latency and token cost; RIG combined with FVES can perform strongly in some cases, but suffers from instability due to judgment errors and coordination overhead. In contrast, OHG with TKS achieves near-optimal accuracy with lower latency than GS, providing a robust balance between solution-space coverage, judgment reliability, and execution efficiency. This observation motivates our choice of OHG and TKS as the default configuration.

| Level | Metric | Magentic-One | OHG+FVES | RIG+FVES | OHG+GS | OHG+TKS | RIG+GS | RIG+TKS |
|-------|--------|-------------|----------|----------|--------|---------|--------|---------|
| **L1** | Accuracy (%) | 43.2 | 35.7 | 39.0 | 51.9 (59.3) | 56.5 (60.9) | **56.7** (70.0) | 44.4 (59.3) |
| | Time (s) | 230.0 | 200.5 | **135.7** | 338.4 | 223.9 | 211.6 | 172.7 |
| | Token (k) | 38.8 | 60.1 | 48.4 | 82.1 | 67.6 | 70.9 | 54.1 |
| **L2** | Accuracy (%) | 25.8 | 25.0 | 28.1 | **38.7** (48.4) | 32.0 (44.0) | 23.1 (42.3) | 30.8 (38.5) |
| | Time (s) | 320.0 | 193.8 | **168.8** | 281.5 | 261.5 | 250.1 | 204.4 |
| | Token (k) | 41.4 | 56.8 | 52.2 | 81.8 | 79.2 | 80.9 | 66.0 |
| **L3** | Accuracy (%) | 9.2 | 8.5 | 7.7 | **11.5** (16.2) | 10.0 (15.4) | 10.8 (11.5) | 9.2 (10.0) |
| | Time (s) | 772.2 | 257.9 | **217.9** | 401.2 | 268.4 | 352.4 | 282.5 |
| | Token (k) | 50.4 | 64.0 | 63.0 | 130.9 | 81.7 | 84.0 | 94.9 |

*Table 1.* Results across different GAIA difficulty levels. Accuracy is reported in percentage (%), token usage in thousands (k), and time in seconds (s). Parenthesized values denote Oracle Accuracy, which measures whether at least one replica produces a correct answer before final selection. The best and second-best values are highlighted with green and blue, respectively (only for the maximum Accuracy and minimum Time within each level).

**OHG exhibits a smaller gap between practical and oracle performance.** Across all difficulty levels, OHG-based configurations show a consistently smaller gap between actual accuracy and Oracle Accuracy compared to RIG-based configurations, indicating more stable and predictable behavior during the selection phase. We hypothesize that OHG introduces more discriminative solution perspectives during parallel generation, thereby reducing the decision burden on the judge. This further justifies our choice of OHG as the default generation strategy.

**Robust judge mechanisms are critical for unlocking the benefits of parallelism.** Regardless of whether GS, TKS, or FVES is used, actual accuracy remains consistently below the corresponding Oracle Accuracy across tasks and configurations. This persistent gap indicates that judge quality plays a decisive role in determining final system performance. Thus, the benefit of parallel exploration depends not only on generating correct trajectories, but also on reliably judging, comparing, and selecting among multiple candidate outputs produced by parallel execution.

**Structural parallelism significantly reduces latency, especially for harder tasks.** On Level 1 tasks, some parallel configurations already outperform Magentic-One in execution time (230s), though the margin is limited. Starting from Level 2, latency reductions become more pronounced: Magentic-One requires approximately 320s, while parallel configurations reduce this to around 170–280s. This trend further amplifies on Level 3, suggesting that as task complexity increases, more internal parallelism is exposed and Structural Parallelism increasingly compresses critical execution paths. Appendix B provides critical-path and parallelizable-node statistics supporting this interpretation.

**The synergy of two-tier parallelism is most pronounced for medium-difficulty tasks.** Compared to Level 1, parallel execution on Level 2 tasks yields substantial latency reduction (from ∼320s to 170–280s), indicating effective Structural Parallelism. Compared to Level 3, Level 2 tasks exhibit larger accuracy gains (from ∼26% to up to 39%), suggesting that parallel generation and selection can still effectively translate expanded solution-space coverage into performance improvements. Overall, medium-difficulty tasks represent a "golden stage" where Structural Parallelism is well-exploited and Replica Parallelism remains effective, leading to the strongest synergistic effects.

### 4.3. Fine-grained Analysis

**Ablation Study.** To verify the independent effectiveness of the two parallelism mechanisms, we conduct ablation studies on Replica Parallelism and Structural Parallelism. Disabling Replica Parallelism sets the number of solution instances to $n = 1$, while disabling Structural Parallelism enforces Strict Sequential scheduling. Following §4.1, we fix OHG and TKS as default settings. The results are shown in Figure 3. Multiple solution instances consistently improve accuracy across difficulty levels. With Structural Parallelism fixed to Balanced Parallelism, increasing $n$ from 1 to 3 or 5 yields higher accuracy on both Level 1 and Level 2 tasks, showing that multiple complete solution paths improve solution-space coverage. Structural Parallelism effectively reduces end-to-end latency. With $n = 3$, switching from SS to BP or AP significantly shortens execution time. However, accuracy drops notably under AP, as analyzed below. Overall, Replica Parallelism improves accuracy while Structural Parallelism reduces latency, and the two complement each other under different constraints.

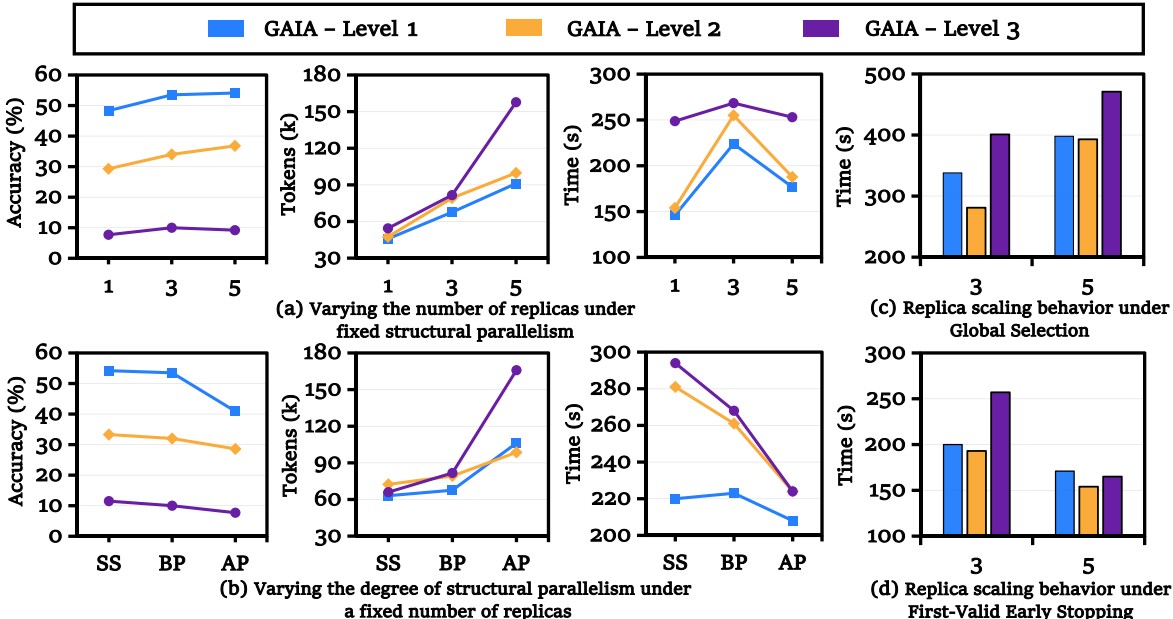

*Figure 3.* Results on how Replica Parallelism and Structural Parallelism affect accuracy and execution time across different task complexities.

| Structural Policy | Replica = 1 | | | Replica = 3 | | | Replica = 5 | | |
|---|---|---|---|---|---|---|---|---|---|
| | Acc. (%) | Time (s) | Token (k) | Acc. (%) | Time (s) | Token (k) | Acc. (%) | Time (s) | Token (k) |
| SS | 29 | 164 | 51 | 33 (37) | 281 | 72 | 33 (38) | 207 | 88 |
| BP | 29 | 154 | 57 | 32 (44) | 261 | 79 | 37 (47) | 188 | 99 |
| AP | 28 | 144 | 66 | 29 (31) | 225 | 98 | 25 (34) | 174 | 124 |

*Table 2.* Joint effect of Replica Parallelism and Structural Parallelism on Level-2 tasks under OHG+TKS. Accuracy is reported in percentage (%), time in seconds (s), and token consumption in thousands (k). Parenthesized values denote Oracle Accuracy.

**Parameter Study.** To further understand the roles and applicability of key parameters, we analyze the number of solution instances and the degree of structural parallelism under fixed OHG and TKS settings. Increasing the number of solution instances improves accuracy, but with rapidly diminishing returns. Increasing $n$ from 1 to 3 yields stable accuracy gains, while further increasing to $n = 5$ brings marginal or even negative improvements, despite significantly higher token consumption. We hypothesize that excessive instances introduce overly diverse strategies that dilute the relative proportion of effective solutions. TKS mitigates the latency pressure introduced by more instances. While token consumption continues to grow, end-to-end latency does not increase when $n$ grows from 3 to 5, and may even decrease. In contrast, GS amplifies latency with larger $n$, while FVES reduces latency at the cost of degraded accuracy. Structural parallelism is not always better when more aggressive. Increasing structural parallelism from BP to AP leads to substantial accuracy drops and increased token consumption. Although AP further shortens execution time in some cases, these gains are unstable and often offset by redundant execution and scheduling overhead, resulting in degraded overall performance.

**Interaction Between the Two Tiers.** Table 2 directly varies the number of replicas and the degree of Structural Parallelism on Level-2 tasks, showing that the two tiers interact rather than contribute additively. Structural Parallelism buffers the time cost of replica scaling: as replicas increase, BP and AP mitigate latency overhead more effectively than SS. Meanwhile, accuracy gains from replica scaling are clearly modulated by the structural policy. Under BP, accuracy rises from 29% to 37% as replicas grow, whereas under the more aggressive AP it drops from 28% to 25%, indicating that excessive structural parallelism can undermine or even negate the accuracy gains from replication. Overall, the accuracy benefit of Replica Parallelism is modulated by Structural Parallelism, while the latency benefit of Structural Parallelism shifts with replica scale, demonstrating a significant interaction between the two tiers.

| Task Type | Metric | Combined Parallelism | Replica-Only | Structural-Only |
|---|---|---|---|---|
| Web | Acc. (%) | 27.12 | 19.08 | 21.60 |
| | Time (s) | 255 | 323 | 172 |
| | Token (k) | 67.7 | 57.5 | 27.1 |
| Code | Acc. (%) | 28.20 | 25.32 | 32.28 |
| | Time (s) | 165 | 301 | 111 |
| | Token (k) | 88.5 | 48.1 | 24.9 |
| Multimodal | Acc. (%) | 12.60 | 12.00 | 7.08 |
| | Time (s) | 266 | 318 | 169 |
| | Token (k) | 68.4 | 47.3 | 30.6 |
| File | Acc. (%) | 65.04 | 54.60 | 39.96 |
| | Time (s) | 233 | 239 | 124 |
| | Token (k) | 36.7 | 29.3 | 17.6 |

*Table 3.* Task-type breakdown on Level-2 tasks. Accuracy is reported in percentage (%), token usage in thousands (k), and time in seconds (s). Combined Parallelism uses both Replica Parallelism and Structural Parallelism; Replica-Only Parallelism disables structural scheduling; Structural-Only Parallelism uses a single replica with structural scheduling.

## 4.4. Discussion

This section discusses the broader implications of our findings and summarizes what they reveal about the role, behavior, and practical use of inference-time parallelism in LLM-based multi-agent systems.

**The two tiers interact non-additively.** The joint analysis shows that Replica Parallelism and Structural Parallelism are not independent knobs whose benefits simply add together. BP can reduce the latency cost of replica scaling while preserving the accuracy gains from broader solution exploration. AP, however, can undermine those gains by introducing redundant execution or weakening information integration. This interaction is central to the TIPEX perspective: the value of one tier depends on how the other tier is configured.

**Parallelism is regime-dependent.** The benefits of parallel execution vary across task difficulty and task type. Harder tasks expose longer critical paths and more parallelizable structures, which creates more room for Structural Parallelism to reduce latency. At the same time, Table 3 shows that web and file tasks favor combined parallelism, code tasks favor structural decomposition, and multimodal tasks rely more on replica-level exploration. These patterns indicate that practical systems should configure parallelism based on task structure and resource constraints rather than applying a fixed level of concurrency.

**Parameter effects are non-monotonic.** Increasing the number of solution instances improves accuracy, but with rapidly diminishing returns. Increasing $n$ from 1 to 3 yields stable accuracy gains, while further increasing to $n = 5$

brings marginal or even negative improvements in some configurations, despite significantly higher token consumption. TKS mitigates the latency pressure introduced by more instances because it can stop after collecting a reliable subset of candidates. Structural Parallelism is also non-monotonic: moving from BP to AP can shorten execution time, but the resulting gains are unstable and often offset by redundant execution and degraded reasoning quality. Overall, our findings suggest that moderate replica parallelism combined with balanced structural parallelism yields the most robust trade-off among accuracy, latency, and computational cost.

**Toward adaptive parallel execution.** Our main experiments use fixed configurations to keep comparisons interpretable, but the observed regime dependence motivates adaptive scheduling. A preliminary Auto Router study in Appendix D shows that task-aware routing can reduce latency, although it does not yet improve accuracy or token cost. Future work should develop more reliable routing mechanisms that decide replica scale, structural scheduling aggressiveness, and selection strategy from task features and intermediate execution states.

**Generalization beyond GAIA.** We additionally evaluate TIPEX on GAIA2-mini (Froger et al., 2026) using DeepSeek-v3.2 (Liu et al., 2025) as the backbone. The full results are reported in Appendix C. The same qualitative patterns remain: Replica Parallelism expands solution-space coverage, Structural Parallelism compresses wall-clock time. The stronger gains on GAIA2 are accompanied by higher token costs, reinforcing that inference-time parallelism should be understood as a joint accuracy–latency–cost trade-off rather than a free improvement.

## 4.5. Failure Analysis

The gap between actual accuracy and Oracle Accuracy shows that judge quality remains a bottleneck for realizing the benefits of Replica Parallelism. We further categorize 104 failed cases across GAIA L1–L3 to identify where current two-tier parallelism still fails.

- **Reasoning/computation error (64/104):** The system applies incorrect reasoning steps, uses invalid formulas, or makes numerical mistakes despite retrieving relevant information.

- **Information retrieval error (24/104):** The system fails to locate, extract, or verify critical evidence from web pages, files, or multimodal inputs.

- **Error propagation (11/104):** Early mistakes in decomposition, retrieval, or intermediate reasoning are carried forward and affect later agents or subtasks.

- **Redundant execution (5/104):** Parallel branches produce overlapping or weakly complementary outputs, increasing cost without providing additional evidence.

These results indicate that parallelism alone cannot replace stronger reasoning, retrieval, and verification mechanisms in complex agent workflows. They also motivate intermediate validation and better path-diversity controls, especially when aggressive scheduling increases redundant execution or weakens information integration.

## 5. Conclusion

We presented TIPEX, a two-tier perspective on inference-time parallelism for LLM-based multi-agent systems, separating Replica Parallelism across solution paths from Structural Parallelism within a path. By unifying both levels under a controllable execution semantics, TIPEX enables systematic combinations of generation, scheduling, and selection strategies while keeping the underlying agent pipeline unchanged. Experiments on GAIA show that parallelism can improve accuracy and reduce wall-clock latency, but at the cost of higher token consumption. Our analyses highlight complementary effects across task difficulty and reveal clear non-monotonicity when parallelism becomes overly aggressive. These results suggest that practical deployments should treat parallelism as a tunable control knob rather than a maximization target. Taken together, these findings indicate that the benefits of inference-time parallelism are inherently regime-dependent, and that optimal performance emerges from carefully balanced configurations that respect both task structure and coordination overhead, rather than from indiscriminate increases in parallel execution.

## Acknowledgements

We thank the anonymous reviewers for their constructive comments and valuable suggestions. We are grateful to Runze Zhang and Jiaxin Li for their helpful feedback, and to Professors Dong Zhao and Haihong E for their insightful guidance. We also thank the students of the BUPT 101 Topnotch Program for useful discussions, as well as the teachers of the program for their continuous support. This work was supported by the Beijing Natural Science Foundation under Grant Nos. QY25338 and QY25335.

## Impact Statement

This paper presents work whose goal is to advance the field of machine learning. There are many potential societal consequences of our work, none of which we feel must be specifically highlighted here.

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

# A. Additional Method Details

## A.1. Judge Implementation and Overhead

The judge operator $\mathcal{J}$ is used only after replica execution has produced candidate traces. It first removes malformed or failed outputs, then scores each remaining candidate along three dimensions: whether the answer is directly usable, whether the trace provides sufficient evidence, and whether the reasoning is internally consistent. This design separates candidate generation from candidate selection, making it possible to analyze the gap between Oracle Accuracy and final accuracy.

| Dimension | 0 | 1 | 2 |
|---|---|---|---|
| Answer Definiteness | No usable answer | Indirect or incomplete answer | Directly adoptable answer |
| Evidence Strength | No valid evidence | Partial or weak evidence | Clear supporting evidence |
| Reasoning Consistency | Severe contradiction | Mostly consistent with gaps | Fully consistent |

*Table 4.* Scoring rubric used by the judge operator. The rubric favors candidates that provide a directly adoptable answer, explicit supporting evidence, and a consistent reasoning trace.

Table 5 reports the relative cost of judge calls. The judge accounts for roughly 4–5% of total latency and token usage, so it is not the dominant cost source. Its importance is instead qualitative: selection errors can prevent the system from realizing the upper bound indicated by Oracle Accuracy.

| Level | Judge / Total Latency | Judge / Total Tokens |
|---|---|---|
| L1 | 5.03% | 4.56% |
| L2 | 4.29% | 3.90% |
| L3 | 5.07% | 4.29% |

*Table 5.* Judge overhead relative to the total TIPEX pipeline. The dominant cost remains replica execution and structural scheduling, but judge quality directly controls how much of the available candidate quality is converted into the final answer.

## A.2. OHG Diversity Diagnostics

OHG synthesizes multiple strategy instructions from the original task and agent-team context. The term "orthogonal" refers to methodological diversity: strategies are encouraged to differ in solving path, processing approach, verification mechanism, and risk preference. To verify this diversity, we embed each generated strategy instruction with `paraphrase-multilingual-MiniLM-L12-v2` and compute the mean pairwise cosine distance across replicas.

| Level | OHG | RIG |
|---|---|---|
| L1 | 0.322 | 0.005 |
| L2 | 0.303 | 0.010 |
| L3 | 0.294 | 0.008 |

*Table 6.* Mean pairwise cosine distance between generated strategy instructions. Higher values indicate greater semantic diversity. OHG consistently produces more diverse strategy instructions than RIG, supporting its role as a structured replica-generation mechanism rather than simple stochastic resampling.

## A.3. Structural-Parallelism Failure Handling

TIPEX uses local failure handling to prevent one failed node from blocking the whole execution graph. Each node is executed with bounded retries and timeout control. When a node fails, the scheduler classifies whether it is a hard dependency for downstream nodes.

- If a failed node is a hard dependency, the affected dependency chain is terminated because downstream computation would be ill-posed.

- If the failed node is not a hard dependency, the scheduler records a summary of the failure and allows independent branches to continue along the available DAG frontier.

- This mechanism preserves concurrency for independent branches while avoiding silent propagation of invalid intermediate results.

This design is intentionally conservative: it does not attempt to hide all failures, but it prevents local failures from unnecessarily collapsing the entire structural execution graph.

## B. Additional Experimental Diagnostics

### B.1. Structural Parallelism Opportunity

The main text observes that latency reductions become more pronounced on harder GAIA tasks. We quantify this behavior from two complementary perspectives: critical-path compression and the prevalence of parallelizable nodes. The critical path measures the longest dependency-constrained execution chain, while the parallelizable-node ratio measures how frequently the scheduler can expose concurrent work.

| Level | Original Critical Path | Parallelized Critical Path | Compression Ratio |
|-------|------------------------|----------------------------|-------------------|
| L1    | 4.36                   | 2.82                       | 35.32%            |
| L2    | 5.71                   | 3.29                       | 42.38%            |
| L3    | 8.92                   | 4.38                       | 50.90%            |

*Table 7.* Critical-path statistics under structural scheduling. Harder tasks expose longer critical paths and therefore larger compressible margins. This explains why latency improvements become more visible as task difficulty increases.

| Level | Avg. Parallelizable Node Ratio | Tasks with Parallel Structures |
|-------|-------------------------------|-------------------------------|
| L1    | 24%                           | 48.98%                        |
| L2    | 43%                           | 70.73%                        |
| L3    | 64%                           | 81.82%                        |

*Table 8.* Prevalence of parallelizable structures under OHG+TKS, $n = 3$, and BP. The opportunity for Structural Parallelism is not uniformly distributed; harder tasks more frequently expose independent or weakly dependent execution units.

Together, Tables 7 and 8 support the claim that Structural Parallelism is most useful when a task contains both a long compressible path and enough independent work to schedule concurrently.

## C. Generalization Studies

### C.1. GAIA2 Results

We evaluate TIPEX on GAIA2-mini using DeepSeek-v3.2 as the backbone model. Each configuration is repeated three times and averaged. Results in parentheses denote Oracle Accuracy. GAIA2 is more dynamic than the original GAIA setting, so this experiment tests whether the observed trade-offs persist when execution traces become longer and more environment-dependent.

| Policy | Metric     | Replica=1 | Replica=3      | Replica=5      |
|--------|------------|-----------|----------------|----------------|
|        | Acc. (%)   | 7.08      | 9.79 (10.42)   | 12.08 (13.33)  |
| SS     | Time (s)   | 681       | 849            | 1131           |
|        | Token (k)  | 45        | 125            | 214            |
|        | Acc. (%)   | 6.88      | 10.63 (12.29)  | 11.25 (12.71)  |
| BP     | Time (s)   | 244       | 348            | 422            |
|        | Token (k)  | 63        | 190            | 298            |
|        | Acc. (%)   | 6.25      | 8.12 (9.58)    | 11.46 (12.50)  |
| AP     | Time (s)   | 223       | 328            | 464            |
|        | Token (k)  | 72        | 216            | 307            |

*Table 9.* GAIA2-mini results under OHG+TKS. Replica scaling improves accuracy in most settings, BP substantially reduces latency relative to SS, and AP remains non-monotonic. The higher token costs indicate that the accuracy–latency–cost trade-off becomes more salient in dynamic environments.

## C.2. Cross-Backbone Validation

We also run supplementary Level-2 experiments with Gemini-3-Flash Preview as the backbone. The core conclusions remain stable: AP reduces latency but can degrade accuracy, and scaling replicas beyond moderate levels yields diminishing returns with higher token cost.

| Metric | SS | BP | AP |
|---|---|---|---|
| Acc. (%) | 39 (48) | 39 (44) | 37 (43) |
| Token (k) | 58 | 74 | 78 |
| Time (s) | 290 | 245 | 234 |

*Table 10.* Structural Parallelism comparison with Gemini-3-Flash Preview on Level-2 tasks, using OHG+TKS and $n = 3$. BP and AP reduce latency relative to SS, but AP slightly lowers accuracy, matching the non-monotonic behavior observed with the default backbone.

| Metric | $n = 1$ | $n = 3$ | $n = 5$ |
|---|---|---|---|
| Acc. (%) | 33 | 39 (44) | 41 (50) |
| Token (k) | 24 | 74 | 110 |
| Time (s) | 189 | 245 | 223 |

*Table 11.* Replica scaling with Gemini-3-Flash Preview on Level-2 tasks under BP and TKS. Increasing the number of replicas improves Oracle Accuracy, but the accuracy gain from $n = 3$ to $n = 5$ is small compared with the additional token cost.

## D. Sensitivity and Limitations

### D.1. Judge Sensitivity

Judge sensitivity is evaluated by fixing the candidate outputs and varying only the judge model or judge prompt. This isolates the selection component from the generation component. The results show that model choice has a larger effect than modest prompt variation, confirming that judge quality is a substantive bottleneck rather than a negligible implementation detail.

| Judge Model | Relative to Oracle (%) |
|---|---|
| Qwen-Plus | 69.62 |
| Gemini-3-Flash Preview | 78.31 |
| Claude Sonnet 4.6 | 65.93 |

*Table 12.* Judge model sensitivity with task inputs fixed. Higher values indicate that the judge recovers a larger fraction of the available Oracle Accuracy.

| Prompt | Relative to Oracle (%) |
|---|---|
| A (default) | 69.62 |
| B (simplified) | 68.03 |
| C (enhanced) | 71.27 |

*Table 13.* Judge prompt sensitivity with Qwen-Plus fixed. Prompt changes have a smaller but still measurable effect, motivating more robust selection mechanisms.

### D.2. Preliminary Auto Router Study

The main experiments use fixed configurations to preserve interpretability. As a preliminary adaptive extension, Auto Router selects two-tier parallelism parameters from task descriptions and attachments before execution.

| Config | Acc. (%) | Time (s) | Token (k) |
| --- | --- | --- | --- |
| OHG+TKS+BP ($n = 3$) | 32.0 (44.0) | 262 | 79 |
| OHG+TKS+Auto Router | 28.1 (42.3) | 199 | 90 |

*Table 14.* Preliminary Auto Router results. The router reduces latency but does not yet improve accuracy or token cost, so we treat it as evidence of extensibility rather than as a mature adaptive strategy.

