# OpenReview forum: "A Two-Tier Perspective on Inference-Time Parallelism in Multi-Agent LLM Systems"
_ICML.cc/2026/Conference — ICML 2026 regular_

### Official Review · Reviewer_SLiH · 2026-03-12

**Soundness:** 3
**Presentation:** 2
**Significance:** 2
**Originality:** 2
**Overall Recommendation:** 4
**Confidence:** 3

**Summary:**

The authors present an execution framework that integrates both replica parallelism and structural parallelism within agent system execution. Different configurations of these two forms of parallelism are examined through experiments conducted on the GAIA benchmark.

**Compliance With Llm Reviewing Policy:**

Affirmed.

**Final Justification:**

The rebuttal addressed one of my concerns on the interaction between two parallelisms, which seems to be the core research question of this work. While the other concern on the generalization beyond the single GAIA benchmark is not fully addressed, the authors promised to include additional results on other benchmarks in the revision.

Given that, I would like to increase the score from 2 to 3, which I think it is appropriate for the current state of the paper.

[Update] Thank you for the additional exp results on GAIA2, which addressed my prior concern on generalization. I will then increase my score to 4.

**Key Questions For Authors:**

1. **What is the key motivation for unifying replica parallelism and structural parallelism?**

The paper proposes a unified framework that coordinates two forms of inference-time parallelism, but it remains unclear why studying their interrelationship is an important research problem. Could the authors clarify the concrete research questions that this unification enables? For example, is the goal to understand new efficiency–accuracy trade-offs, improve scheduling decisions, or provide a principled framework for designing multi-agent execution strategies?

2. **How generalizable are the empirical findings beyond GAIA?**

The experimental evaluation is conducted only on the GAIA benchmark. Given that GAIA tasks are open-ended and web-based, which may introduce potential benchmark contamination or task-specific artifacts, could the authors comment on how well the findings are expected to generalize to other agent benchmarks or tool-use settings? Are there plans or preliminary results on additional datasets?

3. **How prevalent is structural parallelism in realistic agent workflows?**

The effectiveness of structural parallelism depends on the existence of independent or weakly dependent execution steps. In many practical agent pipelines, however, reasoning steps are sequentially dependent. Could the authors provide empirical statistics or qualitative examples showing how frequently parallelizable structures appear in real trajectories (e.g., proportion of nodes that can be executed concurrently under the DAG scheduler)?

4. **Can the authors clarify the presentation of Oracle Accuracy in Table 1?**

Section 4.2 repeatedly discusses the gap between actual accuracy and Oracle Accuracy. However, Table 1 does not explicitly label Oracle Accuracy values, and the numbers in parentheses are described only as “alternative evaluation results.” Could the authors clarify whether the values in parentheses correspond to Oracle Accuracy, and if so, update the table or caption to make this explicit?

5. **How do replica parallelism and structural parallelism interact in practice?**

Although the paper emphasizes the coordination and synergy between the two forms of parallelism, the empirical analysis largely studies them independently (e.g., varying the number of replicas while fixing scheduling strategies). Could the authors provide additional experiments or analysis exploring their interaction, such as joint scaling studies or trade-offs under a fixed compute budget?

**Limitations:**

yes

**Strengths And Weaknesses:**

## Strengths

1. **Addresses an important and practical problem in LLM-based multi-agent systems**

The paper studies inference-time efficiency in LLM-based multi-agent systems, which is an important and practical challenge for real-world deployments. As these systems often require multiple model calls and complex coordination across agents and tools, improving their efficiency without sacrificing solution quality is a key concern for both research and practice.

2. **Provides a systematic empirical study of parallelism in multi-agent execution**

The paper conducts relatively fine-grained empirical analyses of different forms of parallelism in multi-agent systems, including replica parallelism and structural parallelism. The experimental results explore how these mechanisms affect accuracy, latency, and token consumption under different configurations, providing useful insights into the design space of inference-time parallel execution. While the interaction between the two forms of parallelism could be further examined, the study nevertheless contributes valuable empirical observations about how different parallel strategies behave in practice.

## Weakness

1. **Unclear motivation for unifying the two forms of parallelism**

The paper proposes a unified framework that coordinates replica parallelism and structural parallelism, but it remains unclear why such unification is necessary or what specific research questions this unification enables. While the framework conceptually organizes these two forms of parallelism, the paper does not clearly articulate why understanding their interrelationship is an important research problem. Without a clear problem motivation, it is difficult to assess whether unifying the two parallelism mechanisms represents a substantial conceptual contribution beyond engineering system design.

2. **Limited evaluation on a single benchmark**

The empirical evaluation is conducted solely on the GAIA benchmark. While GAIA is a challenging benchmark for general AI assistants, relying on a single dataset limits the strength and generalizability of the conclusions. Moreover, GAIA tasks are open-ended and heavily rely on web-based information retrieval, which raises potential concerns about benchmark contamination given the widespread availability of GAIA data and related solutions online. Evaluating the proposed framework on additional benchmarks (e.g., other agent-based or tool-use benchmarks) would help strengthen the empirical evidence and demonstrate that the observed benefits are not specific to GAIA.

3. **Unclear prevalence and applicability of structural parallelism in practice**

The effectiveness of structural parallelism depends on the existence of independent or weakly dependent execution steps within an agent trajectory. However, in many practical agent workflows, reasoning steps and tool calls are sequentially dependent, making it unclear how often meaningful structural parallelism actually exists in real tasks. Section 3.3 discusses different scheduling strategies in terms of aggressiveness, but the description remains somewhat abstract and lacks concrete examples or statistics showing how frequently parallelizable structures arise in practice. Providing clearer explanations or empirical measurements of the degree of parallelism in typical agent trajectories would help justify the practical relevance of structural parallelism.

4. **Limited analysis and clarity in the experimental results**

The discussion of the main experimental results in Section 4.2 could benefit from deeper analysis and clearer presentation. For instance, the paper reports that OHG performs better than alternative strategies, but it does not provide sufficient explanation for why this happens; qualitative case studies or error analyses could help better understand the underlying reasons. In addition, Section 4.2 frequently discusses the gap between actual accuracy and Oracle Accuracy, yet Table 1 does not clearly present Oracle Accuracy values (the numbers in parentheses are described only as “alternative evaluation results”), making it difficult to trace the discussion back to the table. Finally, although the paper emphasizes the coordination and synergy between replica parallelism and structural parallelism, the empirical analysis largely studies these two mechanisms independently (e.g., varying the number of replicas while fixing the scheduling strategy, or varying scheduling strategies under a fixed number of replicas). As a result, the experiments do not clearly characterize the interaction or trade-off between the two forms of parallelism. For example, it would be useful to analyze joint scaling behavior (e.g., varying both the number of replicas and the degree of structural parallelism) or explore resource allocation trade-offs between the two under a fixed compute budget.

---

> ### Author Rebuttal · Authors · 2026-03-31
>
> We thank the reviewer for the thorough review and constructive feedback. We appreciate the recognition of the practical relevance of our problem and the systematic experimental analysis. We look forward to further discussion, and address each question below.
>
> ### Q1: Motivation for unifying replica and structural parallelism
>
> Both mechanisms belong to inference-time parallelism yet trade compute for *different* types of gains: replica parallelism primarily improves accuracy, while structural parallelism primarily reduces latency. In practical multi-agent deployments, these gains and their associated costs must be weighed *jointly* rather than optimized independently.
>
> The concrete research goal of this paper is to systematically analyze the accuracy–latency–cost trade-offs in multi-agent execution and understand how different parallelism combinations affect this trade-off. To this end, we propose TIPEX, which models both forms of parallelism as composable execution strategies, enabling systematic comparison and trade-off analysis across configurations.
>
> The significance of this unification is therefore to transform two previously independent optimization levers into a single analyzable execution-decision problem, moving inference-time parallelism from "tune each independently" to "reason about jointly."
>
> ### Q2: Generalizability beyond GAIA
>
> We chose GAIA because it is a *comprehensive* rather than single-capability benchmark: it covers multi-step reasoning, multimodal understanding, web browsing, and tool use, further stratified into three difficulty levels. This provides broad type coverage, allowing us to observe different parallelism strategies under a unified benchmark that closely approximates real-world scenarios.
>
> Regarding contamination, we agree that web-based tasks carry inherent risk, but this is unavoidable in current evaluations of general-purpose agents. We believe this does not materially affect our conclusions, since our analysis focuses on *relative* performance differences across parallelism strategies rather than absolute numbers—relative comparisons remain stable even under minor data biases. Moreover, GAIA is designed to mitigate direct memorization effects, and our manual inspection of experiment logs for web-search tasks revealed no evidence of data leakage.
>
> In the final version and follow-up work, we plan to supplement with complementary benchmarks such as AssistantBench, WebArena, or the recent GAIA-2. We are happy to present additional results during the discussion period if generalizability concerns remain.
>
> ### Q3: Prevalence of structural parallelism in realistic workflows
>
> We agree that not all tasks are naturally amenable to structural parallelism. However, in complex multi-turn tasks for multi-agent systems (e.g., GAIA), conditionally independent sub-tasks—such as parallel information retrieval and tool calls—frequently arise. Once such structures exist, structural parallelism can substantially improve end-to-end efficiency.
>
> We provide statistics on GAIA (OHG+TKS, replica=3, BP) showing how often parallel structures appear:
>
> | Level | Avg. Parallelizable Node Ratio | Tasks with Parallel Structures |
> |:---:|:---:|:---:|
> | L1 | 24% | 48.98% |
> | L2 | 43% | 70.73% |
> | L3 | 64% | 81.82% |
>
> We can provide qualitative case analyses during further discussion and will include detailed examples in the revised appendix.
>
> ### Q4: Presentation of Oracle Accuracy in Table 1
>
> The reviewer's observation is accurate. The parenthesized values indeed correspond to Oracle Accuracy, but our current label "alternative evaluation results" is insufficiently clear. We will explicitly correct the table caption and legend in the final version to align with the main text discussion.
>
> ### Q5: Interaction between replica and structural parallelism
>
> We have conducted joint experiments and present representative Level-2 results below (Oracle Accuracy in parentheses):
>
> | | Replica=1 | Replica=3 | Replica=5 |
> |:---|:---:|:---:|:---:|
> | **SS** acc(%) | 29 | 33 (37) | 33 (38) |
> | **SS** time(s) | 164 | 281 | 207 |
> | **SS** token(k) | 51 | 72 | 88 |
> | **BP** acc(%) | 29 | 32 (44) | 37 (47) |
> | **BP** time(s) | 154 | 261 | 188 |
> | **BP** token(k) | 57 | 79 | 99 |
> | **AP** acc(%) | 28 | 29 (31) | 25 (34) |
> | **AP** time(s) | 144 | 225 | 174 |
> | **AP** token(k) | 66 | 98 | 124 |
>
> Key observations: Structural parallelism buffers the time cost of replica scaling—as replicas increase, BP/AP mitigate latency overhead more effectively than SS. Meanwhile, under BP, accuracy rises from 29% to 37% as replicas grow, whereas under the more aggressive AP it drops to 25%, indicating that excessive structural parallelism can undermine or even negate the accuracy gains from replication. Overall, the accuracy benefit of replica parallelism is modulated by structural parallelism, and the latency benefit of structural parallelism shifts with replica scale—demonstrating a significant interaction between the two.

---

> > ### Author Rebuttal · Reviewer_SLiH · 2026-04-02
> >
> > Thank you for the rebuttal. I will increase my score accordingly.
> >
> > For the revision, I strongly encourage you to include results on additional benchmarks. I still have concerns regarding generalization. If possible, it would be helpful to evaluate on benchmarks that are meaningfully different from GAIA. For example, AssistantBench and WebArena appear relatively similar in nature, whereas GAIA2 or coding-oriented benchmarks such as SWE-bench or Terminal-Bench may provide a more diverse assessment.
> >
> > Additionally, please include results that examine the interaction between replica parallelism and structural parallelism, as this seems to be a central research question of the work.

---

> > > ### Author Response · Authors · 2026-04-07
> > >
> > > We thank you for the thorough evaluation and the constructive comments, particularly regarding the generalizability of GAIA results and the interplay between replica parallelism and structural parallelism. These questions have helped us further clarify the core research objectives and contributions of this paper. We also sincerely appreciate your recognition of our work and the score increase following the previous round of discussion.
> > >
> > > **Supplementary experiments on GAIA2.** We fully understand your concern about generalizability, and have therefore conducted additional experiments on GAIA2 in this latest round. GAIA2 constructs a dynamic application environment simulating real-world digital assistant workflows, comprising 800 core scenarios with a 160-question GAIA2-mini subset for rapid evaluation. We chose GAIA2 precisely because it serves as a direct extension beyond the limitations of the original GAIA: whereas the original GAIA primarily involves tool use, web retrieval, and reasoning in a largely static, read-only setting, GAIA2 extends the evaluation to dynamic, asynchronous, and writable real-world scenarios that require agents to complete tasks in self-evolving environments and handle problems closer to real deployment. GAIA2 is thus more representative of real-world agent execution scenarios in both task design and execution environment, making it well-suited for examining the generalizability of our conclusions in dynamic and complex settings.
> > >
> > > **Experimental setup.** We conducted supplementary evaluations on GAIA2-mini using DeepSeek-v3.2 as the backbone model, with the OHG + TKS configuration. Each experiment was repeated three times and averaged. The results are shown below:
> > >
> > > | | Replica=1 | Replica=3 | Replica=5 |
> > > |---|---|---|---|
> > > | **SS** acc (%) | 7.08 | 9.79 (10.42) | 12.08 (13.33) |
> > > | **SS** time (s) | 681 | 849 | 1131 |
> > > | **SS** token (k) | 45 | 125 | 214 |
> > > | **BP** acc (%) | 6.88 | 10.63 (12.29) | 11.25 (12.71) |
> > > | **BP** time (s) | 244 | 348 | 422 |
> > > | **BP** token (k) | 63 | 190 | 298 |
> > > | **AP** acc (%) | 6.25 | 8.12 (9.58) | 11.46 (12.50) |
> > > | **AP** time (s) | 223 | 328 | 464 |
> > > | **AP** token (k) | 72 | 216 | 307 |
> > >
> > > **Analysis.** These results confirm that the key observations emphasized in our paper remain valid in more complex, realistic environments: replica parallelism primarily expands the solution space and improves success rates, but when used in isolation, it may incur substantial time overhead; structural parallelism primarily compresses the execution critical path and, under appropriate scheduling strategies, can effectively offset the latency overhead introduced by replica scaling, while overly aggressive structural parallelism remains non-monotonically beneficial.
> > >
> > > Notably, we observe that both forms of parallelism yield more pronounced performance gains on GAIA2 compared to GAIA. We believe this is not surprising: tasks in GAIA2 exhibit higher dynamism and execution complexity. In such scenarios, structural parallelism can compress the critical path of longer and more complex execution traces, while replica parallelism can better handle dynamic changes, ambiguity, and uncertainty through multi-path exploration. However, we also observe that these performance gains are accompanied by considerably higher computational costs. The more pronounced gains alongside higher costs on GAIA2 suggest that the performance improvements from inference-time parallelism are fundamentally governed by the trade-off among accuracy, latency, and cost—a trade-off that becomes more salient in more complex execution environments.
> > >
> > > From a practical standpoint, we would like to re-emphasize a point you raised in your original review, which we consider highly important: improving inference efficiency in LLM-based multi-agent systems without significantly sacrificing solution quality is itself a critical and practically relevant problem. Whether for email and calendar management, information retrieval and organization, or more general open-ended agent assistants, real-world deployment cares not only about *whether the task is solved correctly*, but also about *whether it is fast enough, stable enough, and worth the cost*. The supplementary results on GAIA2 further demonstrate that the analyses presented in this paper provide an explanatory perspective for configuring parallelism mechanisms in real-world multi-agent systems across different scenarios.
> > >
> > > We sincerely hope that these supplementary experiments further address your generalizability concerns and strengthen your confidence in our work. We will include more complete GAIA2 results in the revised manuscript and integrate all issues discussed during the review process into the main text or appendix. Should you have any further questions or suggestions, we remain happy to continue refining this work.

---

### Official Review · Reviewer_FC1Y · 2026-03-13

**Soundness:** 3
**Presentation:** 3
**Significance:** 2
**Originality:** 3
**Overall Recommendation:** 3
**Confidence:** 3

**Summary:**

The paper discussed a broad topic, that is inference-time efficiency and accuracy trade-offs in LLM-based multi-agent systems via parallel execution. Overall, a central domain addressed by the paper is how to systematically organize, control, and analyze different forms of inference-time parallelism in multi-agent LLM systems under a unified execution framework. The paper proposes TIPEX, a two-tier abstraction of inference-time parallelism that explicitly separates replica parallelism (task-level solution path exploration) and structural parallelism (intra-solution concurrent execution). The framework is implemented on top of Magentic-One and evaluated on GAIA, with extensive empirical analysis of accuracy, latency, and token cost.

**Compliance With Llm Reviewing Policy:**

Affirmed.

**Final Justification:**

After reading the authors’ rebuttal, I feel that the soundness has improved, and I have raised the corresponding score. Considering both the original paper and the rebuttal, I believe the current overall rating is appropriate.

**Key Questions For Authors:**

As shown in *Weaknesses*.

**Limitations:**

yes

**Strengths And Weaknesses:**

**Strengths**
* S1. The two-tier decomposition of inference-time parallelism is clean, intuitive, and well-motivated.

* S2. The paper does not overclaim theoretical guarantees, and the conclusions are framed empirically.

**Weakness**
* W1. Judge model dependency is underexplored. The paper repeatedly shows that judge quality is the bottleneck, but no sensitivity analysis is provided across judge models or prompts. Since judge error directly caps achievable gains, this deserves deeper discussion or ablation.

* W2. Experimental results rely on a single backbone model (Qwen-Plus). Even a small-scale validation on another model would strengthen claims.

* W3. Key controls such as the number of replicas, scheduling policy, and selection strategy are fixed a priori and do not adapt to task difficulty or intermediate execution states. Given the paper’s own findings on strong regime dependence, this limits the framework’s practical generality.

* W4. While accuracy, latency, and token cost are thoroughly analyzed, the paper provides limited qualitative or granular analysis of where and why parallel executions fail (e.g., error propagation, dependency violations, or redundant reasoning), which could further inform the design principles of two-tier parallelism.

---

> ### Author Rebuttal · Authors · 2026-03-31
>
> We thank the reviewer for the thorough review and constructive suggestions, and look forward to further discussion. Unless noted otherwise, experiments use OHG + TKS, $n{=}3$, BP, each repeated three times and averaged.
>
> ### W1: Judge model dependency
>
> We agree that Judge quality is a critical bottleneck. We now provide a sensitivity analysis: fixing inputs per task, we vary only the Judge model (Qwen-Plus, Gemini-3-Flash Preview, Claude Sonnet 4.6) or the prompt (A: default, B: simplified, C: enhanced rules and consistency constraints, with Qwen-Plus fixed):
>
> | Judge Model | Relative to Oracle (%) |
> |:---|:---:|
> | Qwen-Plus | 69.62 |
> | Gemini-3-Flash Preview | 78.31 |
> | Claude Sonnet 4.6 | 65.93 |
>
> | Prompt | Relative to Oracle (%) |
> |:---|:---:|
> | A (default) | 69.62 |
> | B (simplified) | 68.03 |
> | C (enhanced) | 71.27 |
>
> The results confirm that Judge dependency is real: switching models changes final accuracy, and prompt variation also has a modest effect. More Judge details can be found in our response to Reviewer 1X96 Q1. We will add the sensitivity analysis to the revised appendix.
>
> ### W2: Validation on another backbone model
>
> We present supplementary experiments using **Gemini-3-Flash Preview** on Level-2 tasks, with full results to be included in the revision.
>
> **Structural parallelism comparison** ($n{=}3$):
>
> | Metric | SS | BP | AP |
> |:---|:---:|:---:|:---:|
> | acc (%) | 39 (48) | 39 (44) | 37 (43) |
> | token (k) | 58 | 74 | 78 |
> | time (s) | 290 | 245 | 234 |
>
> **Replica scaling** (BP):
>
> | Metric | $n{=}1$ | $n{=}3$ | $n{=}5$ |
> |:---|:---:|:---:|:---:|
> | acc (%) | 33 | 39 (44) | 41 (50) |
> | token (k) | 24 | 74 | 110 |
> | time (s) | 189 | 245 | 223 |
>
> Although absolute accuracy improves with a stronger backbone, key conclusions hold: (1) AP still shows accuracy degradation vs. BP, confirming excessive parallelism has negative effects; (2) scaling from 3 to 5 replicas yields limited accuracy gains but much higher token cost, indicating non-linear returns. These results further validate our core finding: moderate replica parallelism with balanced structural parallelism achieves a more robust accuracy–latency–cost trade-off.
>
> ### W3: Fixed vs. adaptive controls
>
> We chose fixed parameters for the following reasons. First, our focus is characterizing how parallelism mechanisms affect accuracy, latency, and cost; fixed settings ensure interpretability by avoiding confounds from dynamic strategies. Second, LLM-based dynamic decisions remain unstable in this setting, making reliable comparison hard. Third, our configuration (OHG + TKS) was validated through main experiments to provide a robust balance.
>
> We fully agree that task-adaptive scheduling is an important direction. We introduce a lightweight **Auto Router** module that, before each task, uses an LLM to make structured decisions on two-tier parallelism parameters based on task description and attachments:
>
> | Config | Acc (%) | Time (s) | Token (k) |
> |:---|:---:|:---:|:---:|
> | OHG+TKS+BP ($n{=}3$) | 32.0 (44.0) | 262 | 79 |
> | OHG+TKS+Auto Router | 28.1 (42.3) | 199 | 90 |
>
> Although Auto Router does not yet improve accuracy or token cost, its significant latency reduction demonstrates practical potential for task-adaptive scheduling. TIPEX is extensible toward adaptive strategies; we will discuss this in the revision. See also our task-type breakdown in Reviewer 1X96 Q4 for further motivation.
>
> ### W4: Failure analysis
>
> We systematically categorize 104 failed cases across GAIA L1–L3 (per-level breakdowns will be in the appendix):
>
> | Cat. | Description | Count | Ratio |
> |:---|:---|:---:|:---:|
> | C1 | Reasoning & computation errors | 64 | 61.54% |
> | C2 | Information retrieval failures | 24 | 23.08% |
> | C3 | Error propagation & cascading | 11 | 10.58% |
> | C4 | Redundant parallel execution | 5 | 4.81% |
>
> - **C1**: Agent retrieves relevant info but reasons incorrectly (e.g., extracting wrong character name from a correct Filmweb page, or Judge selecting an incorrect option). Mitigation: intermediate verification and specialized tools.
> - **C2**: Agent cannot access required info (e.g., YouTube login wall blocks all retrieval nodes). Mitigation: expanding external interfaces to reduce channel blocking.
> - **C3**: Single-node errors cascade along DAG dependencies (e.g., wrong initial web result invalidates downstream outputs). Mitigation: intermediate verification to isolate errors early.
> - **C4**: Parallel nodes perform redundant operations (e.g., multiple nodes query the same database and hit the same error). OHG introduces strategy orthogonality but cannot fully guarantee non-overlapping execution—an open problem.
>
> This analysis suggests three improvement directions for two-tier parallelism: (1) intermediate verification and specialized tools to mitigate reasoning errors (C1) and cascading (C3); (2) broader external interfaces to reduce common-mode failures (C2); (3) more effective path orthogonality mechanisms to avoid redundant execution (C4).

---

> > ### Author Rebuttal · Reviewer_FC1Y · 2026-04-02
> >
> > Thank you to the authors for their thoughtful response. I will carefully consider whether to adjust the rating accordingly.

---

> > > ### Author Response · Authors · 2026-04-07
> > >
> > > We thank you for the thorough evaluation and the insightful comments, particularly regarding model generalizability, dynamic scheduling mechanisms, and execution process analysis. These questions have helped us further refine the experimental design and analytical depth of the paper. We also greatly appreciate your recognition of our work—especially your acknowledgment that TIPEX's dual-level parallel architecture is clearly structured, that the empirical analysis is solid, and that the paper avoids excessive theorization. This aligns well with the research positioning we aimed to emphasize.
> > >
> > > Building on this, we would like to provide an additional note: beyond the model generalizability concern you raised, other reviewers have also expressed interest in the generalizability of our findings across different scenarios. To address this, we have supplemented experiments and analyses on GAIA2, as detailed in our discussion with Reviewer SLiH, further validating the stability and applicability of our core conclusions in more dynamic, deployment-realistic settings.
> > >
> > > Overall, in our responses we have provided comprehensive supplementary experiments and analyses addressing the review comments, and have further strengthened the evidence base of the paper in terms of generalizability, practical applicability, and mechanistic interpretation. We believe these additions can further reinforce confidence in the conclusions of this work.
> > >
> > > We are glad that our previous responses have fully resolved your concerns, and we sincerely hope that the above supplements provide sufficient additional reference for your final evaluation.
> > >
> > > We will integrate these supplementary materials into the revised manuscript to further improve the completeness and clarity of the paper. Should you have any remaining questions or suggestions, we remain happy to continue refining this work.

---

### Official Review · Reviewer_1X96 · 2026-03-13

**Soundness:** 3
**Presentation:** 2
**Significance:** 2
**Originality:** 2
**Overall Recommendation:** 4
**Confidence:** 4

**Summary:**

This paper presents TIPEX, a controllable execution framework that models inference-time parallelism in multi-agent LLM systems as a two-tier decision space: (1) Replica Parallelism at the task level, which explores multiple complete solution paths to expand solution-space coverage, and (2) Structural Parallelism at the instance level, which enables concurrent execution within a single solution path through task decomposition and dependency-aware scheduling. The authors conduct comprehensive experiments on the GAIA benchmark, demonstrating that inference-time parallelism can improve accuracy and reduce end-to-end latency, though at the cost of increased token consumption. Key findings include complementary effects between the two parallelism types, with medium-difficulty tasks benefiting most from their coordination.

**Compliance With Llm Reviewing Policy:**

Affirmed.

**Key Questions For Authors:**

- Could you provide more details on the judge operator J? What specific criteria does it use for scoring? Is it a separate LLM call, and if so, what is its latency/token cost contribution?
-  How exactly does the "Strategy Synthesizer" generate orthogonal heterogeneous strategies? What are the specific dimensions of heterogeneity (beyond the examples given), and how do you ensure they are truly orthogonal?
- How does TIPEX handle failures in structural parallelism? If one branch of the DAG fails, does it block dependent nodes, or is there recovery mechanism?
- GAIA covers multiple modalities. Do certain task types (e.g., code execution vs. web browsing) benefit more from replica vs. structural parallelism? A breakdown would be insightful.

**Limitations:**

yes

**Strengths And Weaknesses:**

Strengths

- The two-tier perspective on parallelism (replica vs. structural) provides a clean, systematic way to reason about inference-time execution strategies in multi-agent systems.
- The experiments on GAIA across three difficulty levels provide meaningful insights into how parallelism behaves under varying task complexity.
- The paper honestly reports the accuracy-latency-cost trade-offs, including the non-monotonic behavior where overly aggressive parallelism can degrade performance. This is valuable for practitioners.

Weaknesses

- Evaluation is confined to GAIA. While GAIA is comprehensive, additional benchmarks would strengthen generalizability claims.
- The paper acknowledges that "robust judge mechanisms are critical" but provides limited detail on the judge's architecture, training, or failure modes. The gap between Oracle Accuracy and actual accuracy suggests significant room for improvement here.
- The synthesis process for generating diverse solution strategies is mentioned but not elaborated. What dimensions are used? How is diversity measured and ensured?

---

> ### Author Rebuttal · Authors · 2026-03-31
>
> We thank the reviewer for the careful reading and constructive feedback, and look forward to further discussion. Regarding the GAIA generalizability concern in Weakness 1, we refer to our response to Reviewer SLiH Q2.
>
> Unless noted otherwise, experiments use OHG + TKS, $n{=}3$ replicas, and BP.
>
> ### Q1: Details on Judge $J$
>
> Judge $J$ is a standalone LLM-based evaluator. It first filters replicas for validity, then scores each surviving replica via an independent LLM call along multiple dimensions, selecting the one with the highest composite score. The scoring rubric:
>
> | Dimension | 0 | 1 | 2 |
> |:---|:---|:---|:---|
> | Answer Definiteness | No usable final answer | Answer exists but not directly adoptable | Concise, directly adoptable answer |
> | Evidence Strength | No valid evidence or disconnected | Partial but weak | Clear chain supporting conclusion |
> | Reasoning Consistency | Severe contradiction | Broadly consistent with gaps | Fully consistent |
>
> Judge latency and token overhead relative to the total pipeline:
>
> | Level | Judge / Total Latency | Judge / Total Tokens |
> |:---:|:---:|:---:|
> | L1 | 5.03% | 4.56% |
> | L2 | 4.29% | 3.90% |
> | L3 | 5.07% | 4.29% |
>
> Judge is not a major cost source; the dominant cost comes from replica execution and structural parallelism. However, Judge quality directly affects whether parallelism gains are realized. A sensitivity analysis of Judge model and prompting is provided in our response to Reviewer FC1Y W1. Full implementation details will be added in the revised appendix.
>
> ### Q2: How Strategy Synthesizer generates orthogonal strategies
>
> OHG issues a single meta-reasoning call from the original task to generate $n$ structured strategies, each containing an executable instruction and a solving approach. These are concatenated back with the original task for independent execution, yielding multiple solution paths. We will supplement implementation details and qualitative case analyses in the revision.
>
> Here "orthogonal" means *maximally independent in solving methodology* rather than strictly orthogonal. The Synthesizer's system prompt explicitly specifies four diversity dimensions: **solving path**, **source preference**, **verification method**, and **risk preference**, requiring each strategy to differentiate along these axes.
>
> To verify diversity, we embed each strategy's instruction using paraphrase-multilingual-MiniLM-L12-v2 and compute mean pairwise cosine distances across replicas per question:
>
> | Level | OHG | RIG |
> |:---:|:---:|:---:|
> | L1 | 0.322 | 0.005 |
> | L2 | 0.303 | 0.010 |
> | L3 | 0.294 | 0.008 |
>
> OHG yields substantially higher semantic distances than RIG, confirming stronger methodological diversity.
>
> ### Q3: Failure handling in structural parallelism
>
> TIPEX incorporates explicit failure handling to prevent local failures from blocking execution: (1) Each node uses bounded retries and timeout control to avoid long-tail blocking. (2) Upon failure, nodes are classified by downstream dependency: hard-dependency failures terminate the dependency chain; otherwise the node is marked *ready* with a summary log forwarded downstream. (3) A failed node does not affect independent concurrent branches—the execution graph continues along the DAG frontier.
>
> Failures are thus **local and non-blocking**. Details will be in the revised appendix.
>
> ### Q4: Task-type breakdown of parallelism benefits
>
> Following GAIA Appendix C capability definitions, we multi-label each task (web / multimodal / code / file) and analyze Replica-Only (ROP), Structural-Only (SOP), and Combined Parallelism (CP) on Level-2:
>
> | Task | Metric | CP | ROP | SOP |
> |:---|:---:|:---:|:---:|:---:|
> | web | acc(%) | **27.12** | 19.08 | 21.60 |
> | | time(s) | 255 | 323 | 172 |
> | | tok(k) | 67.7 | 57.5 | 27.1 |
> | code | acc(%) | 28.20 | 25.32 | **32.28** |
> | | time(s) | 165 | 301 | 111 |
> | | tok(k) | 88.5 | 48.1 | 24.9 |
> | multi. | acc(%) | **12.60** | 12.00 | 7.08 |
> | | time(s) | 266 | 318 | 169 |
> | | tok(k) | 68.4 | 47.3 | 30.6 |
> | file | acc(%) | **65.04** | 54.60 | 39.96 |
> | | time(s) | 233 | 239 | 124 |
> | | tok(k) | 36.7 | 29.3 | 17.6 |
>
> Different capability types benefit differently from the two forms of parallelism:
> - **Web**: CP substantially outperforms both ROP and SOP, indicating web tasks need both high-level strategy exploration and low-level parallel evidence gathering.
> - **Code**: SOP achieves the highest accuracy with the lowest latency and tokens, suggesting code tasks are well-suited to structured decomposition and execution.
> - **Multimodal**: CP is marginally better; accuracy is close to ROP, trading tokens for time. Multimodal tasks primarily benefit from multi-path exploration but with limited gains.
> - **File**: CP is optimal with the highest accuracy, balancing time via extra tokens—file tasks benefit from multi-entry retrieval with parallel evidence integration.
>
> Overall, web/file tasks favor CP, code tasks favor SOP, and multimodal tasks rely more on replica exploration.

---

> > ### Author Rebuttal · Reviewer_1X96 · 2026-04-06
> >
> > Thanks the authors for their rebuttal. After carefully considering the response and the discussion with other reviewers, I have decided to maintain my score of Weak Accept.

---

> > > ### Author Response · Authors · 2026-04-07
> > >
> > > We thank you for the thorough evaluation and the insightful comments on our work. We also appreciate that you have followed the discussions with other reviewers. Below, we briefly summarize these discussions and provide further responses to any remaining concerns.
> > >
> > > During the discussion phase, we conducted extensive supplementary experiments and analyses on several core aspects of the paper:
> > >
> > > - In the discussion with **Reviewer 3KNs**, we explained why structural parallelism yields more pronounced latency compression on complex tasks, and clarified the fundamental distinction between aggressive parallelism and speculative execution.
> > > - In the discussion with **Reviewer FC1Y**, we provided additional Judge sensitivity analysis, cross-model validation, a preliminary exploration of adaptive parameter scheduling, and a systematic failure analysis.
> > > - In the discussion with **Reviewer SLiH**, we revealed the interplay between replica parallelism and structural parallelism, and supplemented experiments on the GAIA2 benchmark to strengthen the generalizability of our conclusions.
> > >
> > > We are grateful for your recognition of the systematic two-level parallelism perspective proposed in this paper. In particular, your suggestion to analyze how different task types exhibit distinct preferences for the two forms of parallelism has been highly valuable. Following this suggestion, we conducted a category-wise breakdown by capability type and found significant differences in parallelism preferences across task types. This corroborates the paper's core conclusion that *parallelism benefits exhibit pronounced regime dependence*, and also provides direct motivation for the adaptive parameter scheduling direction we preliminarily explored in the discussion with Reviewer FC1Y. We plan to incorporate this analysis into the revised manuscript and position it as an important direction for framework evolution. We sincerely thank you again for this inspiring suggestion.
> > >
> > > Regarding your initial concern about generalizability beyond a single benchmark, we have supplemented experiments on GAIA2 as an additional testbed in our latest discussion with Reviewer SLiH, further validating the stability and applicability of our core conclusions in more dynamic, deployment-realistic scenarios. We refer you to our latest response to Reviewer SLiH for detailed results, and hope this fully addresses your concern.
> > >
> > > We sincerely hope that the above responses, together with the in-depth discussions with all reviewers, sufficiently resolve your concerns and further strengthen your confidence in our work. We will integrate all supplementary experiments, analyses, and implementation details from the discussion phase into the main text or appendix of the revised manuscript. Should you have any further questions or suggestions, we remain happy to continue refining this work.

---

### Official Review · Reviewer_3KNs · 2026-03-13

**Soundness:** 3
**Presentation:** 3
**Significance:** 2
**Originality:** 2
**Overall Recommendation:** 4
**Confidence:** 3

**Summary:**

TIPEX is a controllable framework that organizes inference-time parallelism for multi-agent systems into two distinct tiers to optimize accuracy and efficiency. Tier-1 applies replica parallelism to explore multiple complete solution paths at the task level, aiming to increase solution-space coverage and robustness. Tier-2 utilizes structural parallelism to decompose a single solution path into concurrent sub-tasks, such as agent interactions and tool calls, to reduce end-to-end latency. Systematic experiments on the GAIA benchmark demonstrate that this two-tier approach significantly improves performance, particularly for medium-difficulty tasks.

**Compliance With Llm Reviewing Policy:**

Affirmed.

**Final Justification:**

The rebuttal addressed my concern. So I maintain my score.

**Key Questions For Authors:**

1. Structural parallelism yields more pronounced latency reductions as task complexity increases (from Level 1 to Level 3). Is this because more difficult tasks inherently have more independent sub-tasks, or simply because longer critical paths provide more opportunities for optimization? Could you provide a more detailed profile?

2. How does the aggressive parallelism policy in TIPEX differ fundamentally from speculative execution in standard LLM inference?

**Limitations:**

1. Prohibitive Resource Cost: The multi-replica approach fundamentally trades computational resources for performance, resulting in a substantial increase in token consumption。

2. Non-Monotonic Returns: Performance does not scale linearly with concurrency; overly aggressive parallel configurations (e.g. Aggressive Parallelism) often introduce redundant overhead and can actually degrade reasoning quality.

**Strengths And Weaknesses:**

Strengths:

1. Structured Parallelism Hierarchy: The paper introduces a two-tier design space that formally distinguishes between replica parallelism at the task level and structural parallelism at the instance level.

2. Performance Gains: Systematic experiments on the GAIA benchmark demonstrate that TIPEX significantly improves accuracy, increasing Level 1 success rates from approximately 43% to up to 57%.

Weaknesses:

1. Computational Cost: Inference time parallelism incurs a substantial increase in token consumptions, trading off computational resources for speed and accuracy.

2. Non-monotonic Performance: Parallel strategies (such as Aggressive Parallelism) do not necessarily yield better performance and can actually degrade reasoning quality and introduce redundant overhead.

---

> ### Author Rebuttal · Authors · 2026-03-31
>
> We thank the reviewer for the careful reading and positive assessment. We appreciate the recognition of TIPEX's two-level parallel structure and convincing results, and look forward to further discussion.
>
> ### Q1: Why does structural parallelism yield larger latency reductions on harder tasks?
>
> We attribute this primarily to **longer critical paths** in more complex tasks, which provide a larger compressible margin for parallel scheduling.
>
> In multi-agent systems, increased task difficulty typically entails longer reasoning chains, more interaction rounds, and more execution steps dependent on intermediate results. Under strictly sequential execution, end-to-end wall-clock latency is dominated by this critical path and grows with task complexity. The core mechanism of structural parallelism is to identify concurrently executable nodes and compress the critical path via parallel dispatch. Consequently, when the original critical path is longer, the achievable wall-clock latency reduction is correspondingly more pronounced. We provide a quantitative breakdown below:
>
> | Level | Original Critical Path Length | Parallelized Critical Path Length | Compression Ratio |
> |:---:|:---:|:---:|:---:|
> | 1 | 4.36 | 2.82 | 35.32% |
> | 2 | 5.71 | 3.29 | 42.38% |
> | 3 | 8.92 | 4.38 | 50.90% |
>
> Meanwhile, we acknowledge that complex tasks also tend to contain more independent sub-tasks, offering additional opportunities for parallel scheduling. However, we argue that the availability of concurrent sub-tasks determines *whether* parallelism can be applied (i.e., a necessary condition), whereas critical path length determines *how much* wall-clock time can be saved. The latter is the principal driver behind the more pronounced latency improvements on harder tasks.
>
> ### Q2: How does aggressive parallelism in TIPEX differ from speculative execution?
>
> This is an excellent question. We will supplement implementation details and execution flow descriptions in the revised appendix to clarify TIPEX's mechanism.
>
> To avoid ambiguity, we first clarify the meaning of **aggressive parallelism (AP)** in our paper: AP operates at the multi-agent execution layer by performing finer-grained task decomposition to enlarge the set of concurrently executable nodes in the execution graph, thereby exposing more parallelism opportunities. In contrast, **speculative execution** is a token-level mechanism within a single model's inference pass, accelerating decoding via a "generate-then-verify-or-rollback" paradigm.
>
> We understand the reviewer's intuition that both mechanisms aggressively expand parallel computation, but they differ fundamentally:
>
> 1. **Granularity of execution**: Speculative execution targets token generation during single-model decoding, whereas AP targets task-level and sub-task-level computation units (e.g., agent calls and tool invocations) in multi-agent workflows.
> 2. **Underlying mechanism**: Speculative execution relies on candidate generation with acceptance/rollback. AP involves no speculation; it emphasizes finer-grained decomposition to maximize the concurrently executable node set in the execution graph.
> 3. **Optimization objective**: Speculative execution accelerates single-model inference, whereas AP restructures task execution paths by maximally exposing latent parallel structures to optimize end-to-end multi-agent efficiency.
>
> In summary, the two operate at different abstraction levels with distinct mechanisms and optimization targets, though both leverage parallelism to unlock speedup potential—the central theme of our study.
>
> ### Additional Remarks on Noted Weaknesses
>
> Regarding **token overhead** and **non-monotonic returns**, we greatly appreciate the reviewer for articulating these points. We emphasize that these observations are among the **core findings** of our paper.
>
> Our results demonstrate that parallelism strategies do not yield cost-free, monotonic performance gains across all dimensions. In multi-agent systems, parallelism benefits are *costly*, *tunable*, and exhibit clear *regime dependence*. The increased token consumption and non-monotonic behavior of aggressive parallelism are precisely the trade-offs revealed by TIPEX as a unified analytical framework. In particular, we observe that moderate replica parallelism combined with balanced structural parallelism and a well-chosen selection strategy (e.g., OHG + TKS) often achieves a more robust balance among accuracy, latency, and token cost—especially on medium-difficulty tasks, which we consider the most representative of real-world workloads.
>
> We believe this understanding of inference-time trade-offs has direct implications for deploying multi-agent systems, as different applications impose different preferences over correctness, latency, and cost. TIPEX's value lies in making such trade-offs **explicit and analyzable**.
>
> We thank the reviewer again for the constructive feedback and will further clarify these points in the final version.

---

> > ### Author Rebuttal · Reviewer_3KNs · 2026-04-02
> >
> > Thanks for addressing my concern. I would like to keep the original score.

---

> > > ### Author Response · Authors · 2026-04-07
> > >
> > > We sincerely thank you for the careful reading of our response and the constructive feedback. We are particularly grateful for the two specific and insightful questions you raised, which have helped us further clarify key design choices and analyses in the paper.
> > >
> > > We also greatly appreciate your recognition of our work—especially your acknowledgment that TIPEX's dual-level parallel architecture is well-organized and that the experimental results are convincing. This positive assessment is very encouraging to us.
> > >
> > > We are glad that our previous response has adequately addressed your concerns. We will further integrate these supplementary clarifications into the revised manuscript to improve the overall clarity and completeness of the paper.
> > >
> > > Should you have any additional questions or suggestions, we welcome further discussion and remain committed to refining this work.

---

### Decision · Program_Chairs · 2026-04-30

**Decision:**

Accept (regular)

**Comment:**

This paper proposes TIPEX, a controllable two-tier execution framework that unifies replica parallelism (exploring multiple complete solution paths at the task level) and structural parallelism (concurrent sub-task execution within a single path via decomposition and dependency-aware scheduling) for inference-time efficiency in multi-agent LLM systems. Systematic experiments on the GAIA benchmark (with supplementary results on the more dynamic GAIA2 in rebuttal) analyze accuracy–latency–token trade-offs under different parallelism configurations. There is one reviewer giving a negative score (original), whose concerns are further addressed by the rebuttal.